# Urinary Collectrin as Promising Biomarker for Acute Kidney Injury in Patients Undergoing Cardiac Surgery

**DOI:** 10.3390/biomedicines11123244

**Published:** 2023-12-07

**Authors:** Johanna Tichy, Sahra Pajenda, Martin H. Bernardi, Ludwig Wagner, Sylvia Ryz, Monika Aiad, Daniela Gerges, Alice Schmidt, Andrea Lassnigg, Harald Herkner, Wolfgang Winnicki

**Affiliations:** 1Department of Anesthesiology, Intensive Care Medicine and Pain Medicine, Division of Cardiac Thoracic Vascular Anesthesia and Intensive Care Medicine, Medical University of Vienna, 1090 Vienna, Austria; johanna.tichy@meduniwien.ac.at (J.T.); sylvia.ryz@meduniwien.ac.at (S.R.); andrea.lassnigg@meduniwien.ac.at (A.L.); 2Department of Internal Medicine III, Division of Nephrology and Dialysis, Medical University of Vienna, 1090 Vienna, Austria; sahra.pajenda@meduniwien.ac.at (S.P.); ludwig.wagner@meduniwien.ac.at (L.W.); monika.aiad@meduniwien.ac.at (M.A.); daniela.gerges@meduniwien.ac.at (D.G.); alice.schmidt@meduniwien.ac.at (A.S.); wolfgang.winnicki@meduniwien.ac.at (W.W.); 3Department of Emergency Medicine, Medical University of Vienna, 1090 Vienna, Austria; harald.herkner@meduniwien.ac.at

**Keywords:** acute kidney injury, biomarker, cardiac surgery, collectrin

## Abstract

Background: Early detection of acute kidney injury (AKI) is crucial for timely intervention and improved patient outcomes after cardiac surgery. This study aimed to evaluate the potential of urinary collectrin as a novel biomarker for AKI in this patient population. Methods: In this prospective, observational cohort study, 63 patients undergoing elective cardiac surgery with cardiopulmonary bypass (CPB) were studied at the Medical University of Vienna between 2016 and 2018. We collected urine samples prospectively at four perioperative time points, and urinary collectrin was measured using an enzyme-linked immunosorbent assay. Patients were divided into two groups, AKI and non-AKI, defined by Kidney Disease: Improving Global Outcomes Guidelines, and differences between groups were analyzed. Results: Postoperative AKI was found in 19 (30%) patients. Urine sample analysis revealed an inverse correlation between urinary collectrin and creatinine and AKI stages, as well as significant changes in collectrin levels during the perioperative course. Baseline collectrin levels were 5050 ± 3294 pg/mL, decreased after the start of CPB, reached their nadir at the end of surgery, and began to recover slightly on postoperative day (POD) 1. The most effective timepoint for distinguishing between AKI and non-AKI patients based on collectrin levels was POD 1, with collectrin levels of 2190 ± 3728 pg/mL in AKI patients and 3768 ± 3435 pg/mL in non-AKI patients (*p* = 0.01). Conclusions: Urinary collectrin shows promise as a novel biomarker for the early detection of AKI in patients undergoing cardiac surgery on CPB. Its dynamic changes throughout the perioperative period, especially on POD 1, provide valuable insights for timely diagnosis and intervention. Further research and validation studies are needed to confirm its clinical usefulness and potential impact on patient outcomes.

## 1. Introduction

Acute kidney injury (AKI) affects a significant proportion of critically ill patients and is associated with increased overall morbidity and mortality [1,2,3,4,5]. Furthermore, the terms subclinical AKI and clinically manifest AKI have become established terms in internal medicine and critical care medicine [5,6,7,8]. The emerging number of identified AKI-specific biomarkers has contributed significantly to this development [9,10,11]. Among others, a bedside test measuring tissue inhibitor of metalloproteinases (TIMP) 2 and insulin-like growth factor-binding protein (IGFBP) 7 has made a substantial clinical contribution [12,13]. Moreover, many other parameters, such as kidney injury molecule (KIM)-1 [14,15], neutrophil gelatinase-associated lipocalin (NGAL) [16], interleukin (IL)-18 [17], neprilysin [18,19], and proenkephalin [20], have been investigated, which provide insight into the condition of renal tubular epithelial cells. Thereby, the cells of the proximal tubule are of particular interest because they represent the most vulnerable site of the nephron due to their high energy turnover to perform multiple functions, such as reabsorption, secretion, and control of urinary pH [21,22]. Thus, under conditions of cardiopulmonary bypass (CPB) or surgical trauma, AKI frequently occurs, primarily affecting the epithelial cells of the proximal tubule due to changes in blood flow in the renal cortex leading to damage of these specific cells [23,24,25,26].

Accordingly, it has been shown in previous work that after cardiac surgery, the measurement of urinary neprilysin concentration can predict AKI [19]. Neprilysin is a protein of the proximal tubule. Likewise, collectrin (also named transmembrane protein (TMEM) 27), a type 1 transmembrane protein, is located at the same anatomical site. It is expressed mainly in the cilia and exhibits heterodimerization with amino acid transporters [27]. The absence of this protein in knockout mice is associated with the impaired excretion of excess amino acids. In a recent study of a cohort of patients with AKI due to various causes, collectrin was found to be a robust indicator of tubule cell damage. Furthermore, this marker protein showed a fundamental difference in the dynamics of its urine concentration compared with all other previously established biomarkers. While KIM1, IL-18, NGAL, TIMP2, and IGFBP7 show increased urinary concentrations in tubular injury at clinical or subclinical stages, urinary collectrin has been found to decrease in AKI [28].

In this study, the urine concentration of collectrin was measured in a patient population at particular risk for developing AKI, i.e., patients undergoing cardiac surgery on CPB. Urinary collectrin was analyzed at four distinct peri-interventional timepoints to assess the value of urinary collectrin as a biomarker for early detection and diagnosis of AKI.

## 2. Materials and Methods

### 2.1. Study Design and Population

This study is a subanalysis of 63 patients from a prospective single-center trial undergoing cardiac surgery on CPB [19]. The study was performed from 30 October 2016 to 25 January 2018. Exclusion criteria for the study included patients under 18 years of age, pregnancy, patients with renal replacement therapy prior to surgery, lack of written informed consent, and medication containing sacubitril. In addition, patients who underwent emergency surgery, pulmonary thromboendarterectomy, heart transplantation, or elective cardiac assist device implantation were excluded. Furthermore, patients with missing or completely frozen samples of investigated timepoints were excluded from the analysis.

We analyzed urine samples in the perioperative period, defined as the time span from preoperative phase (admission to the operation room), intraoperative phase (from start of the operation to leaving the operation room), and postoperative phase (entering and stay in the intensive care unit).

### 2.2. Urine Collection

Following anesthesia induction, a Foley catheter was placed, and urine samples were obtained out of the collection chamber at specific timepoints:Baseline: 60 min after anesthesia induction, before skin incision;Thirty minutes after initiating cardiopulmonary bypass (CPB);End of surgery;Postoperative day (POD) one (i.e., 6:00 am the day after surgery).

The collected urine samples were spun at 3000 RPM for 10 min, and the supernatants were stored frozen in separate aliquots at −80 °C and further processed following a predetermined protocol at the Biobank of the Medical University of Vienna. For each measurement, aliquots were thawed under air flow and immediately used for analysis according to the specific enzyme-linked immunosorbent assay (ELISA) method.

### 2.3. Collectrin (TMEM27) ELISA

Collectrin or TMEM27 was measured using the Cusabio CSB-EL023823HU human Collectrin ELISA kit (CUSABIO, Houston, TX, USA). Urine samples were prediluted in PBS immediately after thawing (at a ratio of 1:10). Fifty µL of the prediluted urine sample was applied in duplicate to each well parallel to the standard. Subsequently, 50 µL of HRP-conjugated detection reagent was added to each well. The sealed plate was then incubated at 37 °C for 60 min. The plate was then washed four times with the wash buffer provided in the assay kit using an ELISA plate washer. After addition of substrate/chromogen mixture A and B (50 mL each), the plate was incubated at 37 °C for 10 min. After addition of 50 µL of stop solution, the plate was read at 450 nm in an ELISA reader and sample concentrations were calculated from the standard curve included with each test plate.

### 2.4. AKI Diagnosis

We classified our patients according to Kidney Disease: Improving Global Outcomes Guidelines (KDIGO) stages [17]. Serum creatinine concentrations were measured preoperatively, at the end of surgery, and on PODs 1 to 7, consecutively, in a certified laboratory using the Jaffe method on an Olympus AU5400 (Olympus America Inc., Center Valley, PA, USA).

### 2.5. Data Processing and Statistical Analyses

We analyzed the data by presenting categorized data as absolute counts with relative frequencies and continuous data as means with standard deviation or median with interquartile range (IQR). To compare baseline variables between study groups, we used the Student’s *t*-test for normally distributed continuous variables, the Mann–Whitney *U* test for non-normally distributed continuous variables, and the χ^2^-test for categorical variables. 

We investigated the association of collectrin and AKI using quantile regression with collectrin as the dependent variable and AKI as the covariable of interest. For multivariable quantile regression analyses, we included gender, age, baseline creatinine, and duration of extracorporeal circulation as additional covariables. We reported the estimates of the regression analyses as the coefficients with 95% confidence intervals and corresponding *p*-values. We performed data management and analysis using Microsoft Excel (Redmond, WA, USA), Stata (version 17, StataCorp., College Station, TX, USA), and GraphPad Prism (version 8, GraphPad Software Inc., Boston, MA, USA). All tests were two-sided, and we considered *p*-values less than 0.05 as statistically significant.

## 3. Results

### 3.1. Study Population

In the original study population of this prospective observational study, we included 100 patients, and 4 of those dropped out as described previously [19]. Additionally, 33 patients were excluded because of missing samples. Finally, 63 patients undergoing cardiac surgery on CPB with a mean age of 67.1 ± 11.6 years were included in the analysis. The demographic characteristics of the study population are shown in Table 1.

In total, 37 patients underwent valve replacement, 8 required coronary artery bypass graft (CABG) surgery, and 18 had a combined surgery of CABG and valve replacement. In 11 patients, it was a redo surgery. The median time of anesthesia was 389 min (IQR 328–456), and the median duration of CPB was 138 min (IQR 108–187). All patients were treated at the same intensive care unit (ICU) after surgery and were subsequently transferred to an intermediate care or normal care unit. During the inpatient stay, blood and urine parameters and hemodynamics were monitored. 

Nineteen patients, corresponding to 30% of the study population, developed postoperative AKI. Seventeen patients had AKI stage I, and two patients had AKI stage II in the postoperative phase. None of the patients required renal replacement therapy. The surgical and clinical features of the study population are shown in Table 1.

### 3.2. Urinary Collectrin Levels during Perioperative Course

An analysis of urine samples showed significant dynamics of collectrin levels during the perioperative course. The baseline collectrin level was 5050 ± 3294 pg/mL, but significantly decreased to 2947 ± 2419 pg/mL 30 min after the start of CPB and reached its nadir at 1438 ± 1795 pg/mL at the end of surgery. On POD 1, collectrin levels began to recover at 3301 ± 3564 pg/mL (Figure 1). 

In the postoperative period, 19 of 63 patients developed AKI according to the KDIGO criteria [29], corresponding to 30% of the study population. The most effective timepoint for distinguishing between AKI and no AKI based on urinary collectrin was on POD 1 with a significant difference in collectrin levels of 2190 ± 3728 pg/mL in AKI patients and 3768 ± 3435 pg/mL in non-AKI patients (*p* = 0.01) (Figure 2).

However, a multiple quantile regression analysis revealed no association between AKI and urinary collectrin on the first postoperative day and at other postoperative timepoints after adjustment for age, sex, extracorporeal circulation time, and baseline serum creatinine (Table 2).

## 4. Discussion

The main focus of this study was to investigate AKI in patients undergoing cardiac surgery on CPB. This specific patient population is particularly vulnerable to develop AKI during the perioperative phase. The particular aim was to assess the concentration of a sensitive parameter called collectrin excreted in urine at an individual patient-specific level. Urinary collectrin, unlike other well-established biomarkers of AKI that increase when tubule cells are damaged, actually decreases in urinary concentration during AKI. This research demonstrates that the urinary collectrin concentration exhibits significant changes throughout the perioperative course, reaching its nadir at the end of the surgical procedure and begins to recover on POD 1.

Due to rapid changes in hemodynamics, proinflammatory stimuli, oxidative stress, hemolysis, myoglobinemia, and organ cross-talk [23,30], cardiac surgery is frequently associated with AKI [31,32,33,34,35]. A specific term to describe this context is cardiac surgery-associated AKI (CSA-AKI), also classified as cardiorenal syndrome type 1, which reflects an acute limitation of cardiac function followed by AKI [25,30,36]. Until now, KDIGO criteria have served as the standard for defining AKI and its stages, based on measurements of serum creatinine and urinary excretion [29]. However, recent research findings have prompted discussions about adding the new term “subclinical/preclinical AKI” to the existing AKI terminology as several urinary AKI biomarkers have been discovered that predict AKI earlier, before detectable changes in serum creatinine occur [6,18,19,37,38]. 

The complex cellular structure of a nephron, combined with the potential for differential damage at the subcellular level in AKI, underscores the significance of investigating different urinary marker proteins to detect preclinical cellular damage. This might be beneficial in the diagnosis and research of AKI of different etiology [9,39].

In our recent study, urinary collectrin was identified as a sensitive and novel biomarker for AKI with unique properties compared to previously identified factors—notably, its urine concentration is reduced during AKI [28]. Consistently, collectrin correlated inversely with serum creatinine and AKI stages. Collectrin is highly specific for proximal tubular cells, with moderate enrichment in the collecting duct (http://nephrocell.miktmc.org/ (accessed on 1 December 2023)) [40,41]. 

The cleavage of the extracellular part of this transmembrane protein, facilitated by a specific protease, is crucial for the release of its water-soluble component into urine. This process shows similarities with physiological mechanisms observed in the islets of Langerhans or ß-cell lines [42]. It is reasonable that this mechanism depends on the proper movement of the brush border or the transport of collectrin to the cilia [43]. The process is energy-dependent, and under conditions of decreased renal blood flow and diminished oxygenation, the proximal tubule may experience energy depletion. This underscores the vulnerability of the proximal tubule to changing hemodynamic states [44].

In this study involving patients undergoing cardiac surgery on CPB, urine collectrin emerged as a valuable biomarker for AKI. An inverse correlation of urinary collectrin concentrations with serum creatinine levels and AKI stages was found. Significant changes in urinary collectrin levels were observed during the perioperative period, with a decrease detected as early as 30 min after the start of CPB. On POD 1, we determined the optimal time for identification of AKI based on urinary collectrin concentrations. The initial decline in urinary collectrin during cardiac surgery is indicative of renal injury, whereas the subsequent increase in collectrin levels in the postoperative period may suggest proximal tubular regeneration, potentially offering insights into the recovery of renal function. Of note, this study does not provide a definitive answer to this question, as further investigation is needed to gain a more comprehensive understanding. The number of studies focusing on biomarkers for predicting renal recovery is limited, with only few studies exploring, i.e., urinary TIMP-2, IGFBP7, and NGAL in relation to renal function recovery after AKI following cardiac surgery [45,46]. While some studies have detected an association between urinary markers and recovery of renal function [47,48,49,50], the prognostic significance of collectrin in this context is still unknown.

The main limitation of the study is its short observation period for the study population. Furthermore, it lacks the capacity to draw conclusions about the potential impact of invasive procedures and anesthetic techniques on the course of collectrin. The study focused exclusively on biomarkers of AKI and relied on AKI diagnosis according to the KDIGO criteria. Incorporating additional biomarkers in serum or urine may have provided a more comprehensive understanding, especially for subclinical AKI. On the other hand, the strength of the study lies in its well-characterized and exclusive study population within a prospectively designed cohort study, leading to robust results.

## 5. Conclusions

Our research identifies urinary collectrin as an innovative biomarker for the detection of AKI in patients undergoing cardiac surgery on CPB. Unlike other established AKI biomarkers, a decline in urinary collectrin reliably signals imminent renal failure. Monitoring collectrin dynamics throughout the perioperative phase enables early detection and timely intervention in AKI cases. 

This work sets the groundwork for deeper exploration of AKI diagnostics and interventions. Nevertheless, the assessment of the clinical utility of collectrin necessitates further research to fully elucidate its impact on patient outcomes.

## Figures and Tables

**Figure 1 biomedicines-11-03244-f001:**
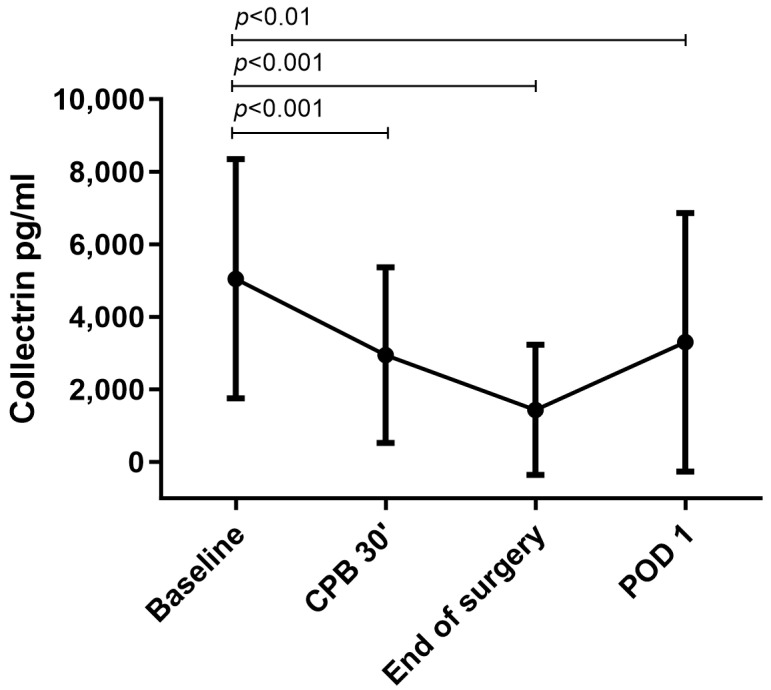
Urinary collectrin levels in the perioperative course of cardiac surgery with CPB. The line graph with error bars shows the perioperative course of urinary collectrin at different time points. Significant differences between the timepoints are indicated above the line graph with *p*-values. Abbreviations: CPB, cardiopulmonary bypass; POD, postoperative day.

**Figure 2 biomedicines-11-03244-f002:**
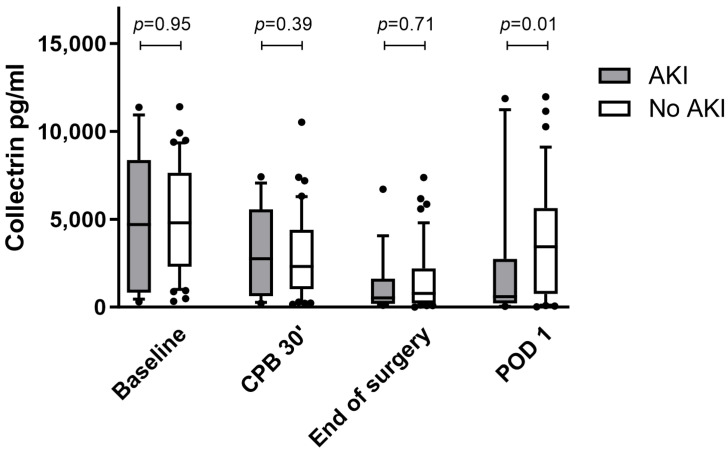
Urinary collectrin levels in patients with and without AKI in the perioperative time course. The boxplots show the difference between patients with (grey) or without (white) postoperative AKI at different time points in the perioperative course. Significant differences between the groups are indicated above the boxplots with *p*-values. In the boxplots, the lower boundary of the box indicates the 25th percentile, a black line within the box marks the median, and the upper boundary of the box indicates the 75th percentile. Whiskers above and below the box indicate the 10th and 90th percentiles. Points above and below the whiskers indicate outliers outside the 10th and 90th percentiles. Abbreviation*s:* AKI, acute kidney injury; CPB, cardiopulmonary bypass; POD, postoperative day.

**Table 1 biomedicines-11-03244-t001:** Demographic and surgical characteristics.

	AKI (N = 19)	No AKI (N = 44)	*p*-Value
Age (years)	70.0 [61.0; 75.0]	69.0 [55.8; 76.0]	0.759
Male	13 (68.4%)	25 (56.8%)	0.560
Female	6 (31.6%)	19 (43.2%)	
BMI	28.0 ± 6.1	26.5 ± 4.7	0.36
Baseline SCr (mg/dL)	0.90 [0.79; 1.04]	0.86 [0.71; 0.97]	0.467
**Comorbidities**			
Asthma	2 (10.5%)	1 (2.27%)	0.214
COPD	3 (15.8%)	6 (13.6%)	0.999
NIDDM	3 (15.8%)	5 (11.4%)	0.688
IDDM	0 (0.00%)	4 (9.09%)	0.306
Chronic kidney disease	2 (10.5%)	2 (4.55%)	0.578
Cardiac decompensation	0 (0.00%)	1 (2.27%)	0.999
PAOD	1 (5.26%)	3 (6.98%)	0.999
Angina pectoris			
Absent	14 (73.7%)	30 (69.8%)	0.999
Stable	5 (26.3%)	11 (25.6%)	
Unstable	0 (0.00%)	2 (4.65%)	
LVEF			
>50%	12 (66.7%)	30 (68.2%)	0.882
30–50%	6 (33.3%)	12 (27.3%)	
<30%	0 (0.00%)	2 (4.55%)	
**Procedure**			
CABG	2 (10.5%)	6 (13.6%)	0.999
Valve	11 (57.9%)	26 (59.1%)	
Combined	6 (31.6%)	12 (27.3%)	
**Surgical characteristics**			
Anesthesia duration (minutes)	376 [322; 450]	389 [332; 457]	0.747
Surgery (minutes)	309 ± 94.6	303 ± 74.6	0.830
CPB (minutes)	148 [112; 188]	135 [106; 179]	0.782
AoCC (minutes)	89.6 ± 43.9	99.1 ± 43.3	0.435
Reoperation	2 (10.5%)	9 (20.5%)	0.480
Crystalloids (mL)	5000 [3750; 5525]	4225 [3500; 5125]	0.392
Intraoperative urinary output (mL)	403 [403; 628]	403 [403; 672]	0.445
Balance intraoperative (mL)	4718 [3510; 5936]	4458 [3704; 5922]	0.999
PRBC (units)	0.00 [0.00; 1.00]	0.00 [0.00; 1.00]	0.874
Platelets (received)	4 (21.1%)	8 (18.2%)	0.999
Fresh frozen plasma (received)	3 (15.8%)	3 (6.8)	0.5185
Fibrinogen (g)	0.00 [0.00; 2.00]	0.00 [0.00; 2.00]	0.724
Coagulation factors (I.U.)	0.00 [0.00; 131.6]	0.00 [0.00; 204.5]	0.6224
**Postoperative complications**			
SAPS 3	45.7 ± 10.8	42.5 ± 7.6	0.2513
no AKI	0 (0.00%)	44 (100%)	<0.001
AKI KDIGO Stage 1	17 (89.5%)	0 (0.00%)	
AKI KDIGO Stage 2	2 (10.5%)	0 (0.00%)	
AKI KDIGO Stage 3	0 (0.00%)	0 (0.00%)	
Renal replacement therapy	0 (0.00%)	0 (0.00%)	0.999
Length of stay on ICU (days)	1.00 [1.00; 5.00]	2.00 [1.00; 3.00]	0.956

Values are presented as number (n) and percentage (%) or median (interquartile range). The listed *p*-values of statistical tests were calculated by using the Student’s *t*-test for normally distributed continuous variables, the Mann–Whitney *U* test for non-normally distributed continuous variables, and the χ^2^ test for categorical variables. Abbreviations: AKI, acute kidney injury; AoCC, aortic cross-clamp; CABG, coronary artery bypass graft; COPD, chronic obstructive pulmonary disease; CPB, cardiopulmonary bypass; ICU, intensive care unit; IDDM, insulin-dependent diabetes mellitus; KDIGO, Kidney Disease—Improving Global Outcomes; LVEF, left ventricular ejection fraction; NIDDM, non-insulin-dependent diabetes mellitus; PAOD, peripheral artery occlusive disease; PRBC, packed red blood cells; SAPS, Simplified Acute Physiology Score; SCr, serum creatinine.

**Table 2 biomedicines-11-03244-t002:** Association between AKI and urinary collectrin adjusted for clinical variables on first postoperative day.

Variable	Coefficient (95% CI) Units Collectrin Change	*p*-Value
AKI crude	−3058 (−5411 to −705)	0.01
AKI-adjusted	−1567 (−4401 to 1266)	0.27
Age (years)	−86 (−214 to 42)	0.18
Sex (male)	−699 (−3371 to 1973)	0.60
Extracorporeal circulation time (per minute)	16 (−7 to 38)	0.17
Baseline serum creatinine (per mg/dL)	−635 (−5161 to 3891)	0.78

Multivariable adjustment for age (years), sex (male/female), extracorporeal circulation time (minutes), and baseline serum creatinine (mg/dL). CI, confidence interval.

## Data Availability

The data that support the findings of this study are available in anonymized form from the corresponding author on reasonable request and after agreement with the local ethics committee.

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
