# Peer review of "Urinary Collectrin as Promising Biomarker for Acute Kidney Injury in Patients Undergoing Cardiac Surgery"

_biomedicines, 2023, doi:10.3390/biomedicines11123244_

Round 1

Reviewer 1 Report

Comments and Suggestions for Authors

In this prospective, observational study, the authors examined the urinary collectrin concentrations of cardiac surgery patients at different time points during the perioperative period, and also followed up to determine whether acute kidney injury (AKI) occurred after surgery. It was found that there was a significant difference in urinary collectrin levels on postoperative day 1 between patients with AKI and those without AKI. This study provided a new potential urine biomarker for predicting postoperative AKI and demonstrated some level of innovation. However, the overall presentation of the results in the manuscript is overly simplified, and there are still many major issues that need to be addressed (see below).

Major Concerns:

1.     Figure 1 illustrates the variations of the same parameter (urinary collectrin) at different time points within the same population. Therefore, a line graph with error bars is more suitable than a box plot for representing this type of change.

2.     Page 3, Line 162: “collectrin correlated inversely with serum creatinine and AKI stages”. Which table or figure in the manuscript presents the data supporting this conclusion? Table 2 only illustrates the relationship between urinary collectrin and AKI, without indicating the relationship with creatinine levels.

3.     Quantile regression allows us to examine the relationship between variables at different points of the distribution, so instead of estimating a single coefficient for each independent variable, we estimate multiple coefficients corresponding to different quantiles. However, in Table 2, what is the significance of the author using quantile regression if I did not see coefficients and P-values changing with quantiles of the dependent variable?

Minor Concerns:

1.     Define perioperative period in the Methods section.

2.     Define postoperative day 1 in the Methods section (24 hours after anaesthesia induction? Or end of surgery?)

Comments on the Quality of English Language

The English language is fine.

Author Response

Dear Reviewer! Thank you very much indeed for the motivating and helpful comments which gave us the opportunity to provide a carefully revised manuscript utilizing the suggestions provided by you. Please find our detailed answers in the attached document!

Reviewer 2 Report

Comments and Suggestions for Authors

The authors explored the possible use of collectrin in urine as a new biomarker for acute kidney injury (AKI) in patients undergoing cardiac surgery. The changes of collectrine levels during and after surgery have been identified as well as relationship between collectrine levels in urine and AKI occurrence. According to reported data, the concentration of this marker in urine could be routinely determined using commercially available ELISA kit following the manufacture instructions that are also described in detail in this manuscript. This analysis can be performed in numerous laboratories and there is no need to buy expensive instrumentation for its performance. So, the authors have made an important step forward in finding a reliable marker of AKI.

I have only two suggestions for minor revisions of the manuscript. Please check all abbreviation as it seems that same of them are used without introducing them (without description).  The short description about determination of serum creatinine should be added, as there is no detail in methodology section in current version of the manuscript

Best regards

Author Response

(The authors gave the same response as above.)

Reviewer 3 Report

Comments and Suggestions for Authors

The manuscript sent by your team for publication is correctly structured, it launches an interesting hypothesis that later through the selection of patients, the application of statistical tests, the results obtained and their discussion is largely supported. the scientific value of the manuscript.

It would have been interesting if your study also took into account the values of urea and creatinine determined preoperatively, perioperatively and postoperatively. Perhaps a parallel follow-up of these values compared to collectrin values would give greater credibility to the study and no further studies would be needed to confirm the usefulness of this new proposed biomarker to predict the onset of AKI in the evolution of these patients.

In addition, a research and discussion of the sensitivity and specificity of collectrin as a biomarker of predictability and a larger group of patients would be needed.

It would still be clear if the invasive procedures applied to patients, as well as the anesthetic technique used, have any role in increasing the concentration of collectrin.

Also, the conclusion of your study is too ambiguous, dry and totally undefined, which induces the idea that the authors themselves are unsure of the results obtained.

The bibliography used is rather old, many bibliographic references exceeding 10 years old and some references even over 20 years old

Author Response

(The authors gave the same response as above.)

Round 2

Reviewer 1 Report

Comments and Suggestions for Authors

The authors have solved all my concerns, no more comments.

Reviewer 3 Report

Comments and Suggestions for Authors

The revisions made to your article are sufficient and the answers given to the objections made by me are satisfactory.